# Microstructure Evolution and Nanotribological Properties of Different Heat-Treated AISI 420 Stainless Steels after Proton Irradiation

**DOI:** 10.3390/ma12111736

**Published:** 2019-05-28

**Authors:** L.Y. Dai, G.Y. Niu, M.Z. Ma

**Affiliations:** 1State Key Laboratory of Metastable Materials Science and Technology, Yanshan University, Qinhuangdao 066004, China; Dly@rxpopcorn.com (L.Y.D.); jnxyngy@163.com (G.Y.N.); 2Labthink Instruments Co. Ltd., Jinan 250031, China

**Keywords:** AISI 420 stainless steel, proton irradiation, microstructure, nano-scratch hardness, friction coefficient, wear rate

## Abstract

In this paper, low-energy proton irradiation experiments with different cumulative fluences were performed on samples of AISI 420 stainless steel that were either annealed or tempered at 600 or 700 °C. The effects of the cumulative proton irradiation fluence on the evolution of the microstructure of AISI 420 were studied by transmission electron microscopy (TEM). Scratch tests were performed using a Tribo Indenter nanomechanical tester, in order to investigate the effects of the cumulative fluence on the tribological properties of the AISI 420 stainless steel. The results indicate that the dislocation density of the microstructure near the surface of the AISI 420 stainless steel increases with higher cumulative proton irradiation fluences. Under the same load, the nanoscale friction coefficient and wear rate both decreased with increasing cumulative proton irradiation fluence. This indicates that the surface hardening effect induced by proton irradiation can diminish the nanoscale friction coefficient and wear rate.

## 1. Introduction

AISI 420 martensitic stainless steel has many positive traits such as good mechanical properties, processability, and a low cost–performance ratio. For these reasons, it has been widely used in diverse domains, such as for aerospace structures, nuclear power plants, steam generators, mixer blades, pressure vessels, turbine blades, petroleum machinery, oil and gas valves, and injection plastic molds [1,2,3,4,5,6]. In the aeronautical and aerospace industries, AISI 420 steel is a common structural material for the manufacture of mobile components, such as gears or shafts [7]. Due to the harsh environments of outer space, aerospace structures are often in exposed environments and are subject to several intense environmental factors, such as particle irradiation, temperature fluctuations, and a high vacuum environment. Among these factors, a pressing area of concern is whether proton irradiation has a significant impact on the tribological and wear performance of the aerospace components with relative motion. Therefore, it is of importance to investigate the tribological and wear properties of AISI 420 stainless steel after proton irradiation.

There is a large existing body of work on the tribological and wear properties of AISI 420 stainless steel [8,9,10,11,12,13,14,15,16,17]. Recently, D’Ans et al. [8] studied the tribological and wear behaviors of the AISI 420/Fe_2_B surface layer by ball-disc tribological and wear tests. For cryogenically-treated AISI 420 steel, Prieto et al. [9,10] performed frictional wear tests with grease lubrication and an argon atmosphere. They found that the wear resistance of the cryogenically-treated specimen was enhanced by ~30–90%, compared to that of the conventionally-treated specimen. Angelini et al. [11] performed a low-temperature plasma carburization on AISI 420 steel over a temperature range of 350–500 °C and a holding time of 4–16 h. The tribological and wear behavior of AISI 420 steel was investigated using the block-on-ring method. The results indicated that low-temperature plasma carburization can effectively reduce wear; the wear mechanism for the specimen treated at 500 °C was slight oxidative wear. In addition, laser surface cladding [12], surface nitriding [13,14,15], diamond-like surface coatings, [16,17] and other techniques have been used to examine the tribological and wear properties of AISI 420 steel under different conditions. However, there have been only a few reports to date, on the effects of proton irradiation on the tribological and wear properties of steel. While there are several reports on proton-irradiated stainless steel [18,19,20,21,22,23], they mostly focus on the effects of proton irradiation on the microstructures and properties of austenitic or ferritic steels that are used in the nuclear industry. Results on the tribological and wear properties of AISI 420 steel after tempering and proton irradiation are rarely reported. In the literature [24], work has been done on the effects of annealing and tempering at 673 K and 943 K on the sliding friction and wear properties of AISI 420 steel. However, this work did not address the effects of proton irradiation on the tribological and wear properties. Usually, proton irradiation can cause point defects in the crystal near the material’s surface; lattice distortion caused by the crystal defects result in the surface-hardening effect [18,25]. Protons are charged particles, and their penetration ability into materials is dependent on their energy. High-energy protons modify the bulk properties, low-energy protons modify only the surface. The effects of surface hardening induced by proton irradiation on the tribological and wear behavior of materials remains an unstudied area of interest. However, proton irradiation only acts on a thin layer under the surface of AISI 420 stainless steel, with an affected thickness measuring in the tens of micrometers; hence, high-load tribological and wear test methods using spherical-disc type and pin-disc type structures are not suitable. The scratch test method, using a nano-mechanical tester, can accurately measure mechanical parameters such as the friction coefficient, scratch resistance, and dynamic scratch hardness of the micro-zone, or a thin layer on the material’s surface. Therefore, it has been applied as an important tool to study the surface properties of various materials, such as films and coating [26]. Since the radius of curvature of the Berkovich indenter used in the nano-scratch test is in the range of only tens of nanometers, a scratch test with a small amount of pressure can be equivalent to the contact friction under extreme loads. Therefore, the test results can effectively reveal the material’s tribological and wear characteristics.

In this paper, the microstructure evolution of annealed, 600 °C-tempered and 700 °C-tempered AISI 420 stainless steel specimens were studied, using proton irradiation experiments with different cumulative fluences. The hardness, friction coefficient, wear volume, and wear rate were tested through the nano-scratch method using a nanomechanical tester (Tribo Indenter). Finally, the relationships between the microstructure evolution, the cumulative fluence of proton irradiation, and the nanotribological properties were investigated.

## 2. Materials and Methods

### 2.1. Experimental Materials and Thermal Processing Conditions

For the study, a round bar of AISI 420 martensitic stainless steel was used, with a diameter D = 20 mm (manufactured by Fushun Special Steel Co., Ltd., Fushun, China) and a density of 7.75 g·cm^−3^. The composition is shown in Table 1.

Considering that the AISI 420 stainless steel, widely used in engineering, is quenched and tempered, the heat treatment system, based on the national standard GB/T1220-2007 of the People’s Republic of China, is set out in this paper. There were two kinds of heat treatment processes in this work: one is the annealing process where an AISI 420 stainless steel round bar with a diameter of 20 mm was raised to 680 °C at a heating rate of 10 °C/min, then cooled to 350 °C and cooled to room temperature. The second was the tempering process. The AISI 420 stainless steel round bar with a diameter of 20 mm was heated to 980 °C for 1 h at a heating rate of 10 °C/min. After being discharged, it was quickly cooled it to room temperature in the oil, then reheated to 600 °C and 700 °C, heat-cooled for 2.5 h, and cooled to room temperature naturally. The process parameters are shown in Table 2.

The AISI 420 steel was annealed and tempered according to the established annealing and quenching-tempering processes in a vacuum tube-type heat treatment furnace (SK-G06143, Tianjin Central Electric Furnace Co., Ltd., Tianjin, China). The furnace could be heated under an argon atmosphere to a maximum temperature of 1300 °C. A heating rate of 10 °C·min^−1^ was used with a temperature precision of ±1 °C.

### 2.2. Proton Irradiation and Nano-Scratch Test

The proton irradiation was performed by using a proton accelerator attached to the ground-based complex irradiation simulation system at the Harbin Institute of Technology (Harbin, China). During the irradiation process, the specimens were placed in a vacuum of 10^−3^–10^−4^ Pa, and maintained at room temperature [27]. Before the proton irradiation test, the AISI 420 steel after annealing and quenching and tempering was processed into the pellet sample shown in Figure 1, and after surface polishing treatment, the pellets were inserted into an irradiation specimen holder, as shown in Figure 2. The irradiation specimen holder was placed on the table with the irradiation target, so that the exposed surface of the sample was perpendicular to the positive ion proton beam. The parameters and conditions for the proton irradiation experiment are shown in Table 3.

The specimens subject to the proton irradiation treatment were subsequently tested in the nano-scratch module of the Tribo Indenter nanomechanical tester (Hysitron, Inc., Eden Prairie, MN, USA). The nano-scratch module has an atomic force microscopy (AFM) system that can capture scratch images. In the AFM system, the surface characteristics of the substrate were detected via the interactions between the micro indenter and the object to be tested. The nanoscratch test used a Berkovich indenter, and the applied loads were 2000 μN, 3000 μN, and 4000 μN, respectively. The scratch length was 15 μm, and the scratch speed was 0.3 μm/s.

### 2.3 TEM Characterization

A transmission electron microscopy (TEM) sample was prepared by a one-side thinning process performed on the non-irradiated side (back side). For this, the irradiated specimen was ground from the back side with metallographic sandpaper to a thickness of about 30 μm. The specimen was punched into a wafer, having a diameter of 3 mm with a punch. Finally, the TEM sample was obtained by a single-sided ion-thinning perforation method. Imaging was conducted using a transmission electron microscope (JEOL2010, Tokyo, Japan) with an acceleration voltage of 200 kV.

## 3. Results

### 3.1. Microstructure Evolution for AISI 420 Stainless Steels with Different Heat-Treatments after Proton Irradiation

Proton irradiation can lead to variations in a material’s physical and mechanical properties, and in the tissue composition and structure. After proton irradiation, defects are generated in the crystalline material, and they are generally referred to as radiation damage. For metallic materials, the simplest radiation defect is an isolated point defect. This defect is usually composed of an interstitial atom that changes to an off-lattice site and the generated lattice vacancy, also known as a Frenkel defect pair [28].

The TEM images in Figure 3, Figure 4 and Figure 5 show the evolution of the microstructures in the annealed, 600 °C-tempered and 700 °C-tempered AISI 420 stainless steel samples after proton irradiation with different cumulative fluences. It can be seen from Figure 3 that the microstructure in the annealed AISI 420 stainless steel has coarser crystal grains, along with the presence of black precipitates. It can also be seen from Figure 3a that the annealed AISI 420 stainless steel without proton irradiation has a small amount of dislocations. From Figure 3b–d, we see that the dislocation density in the specimen after proton irradiation becomes higher with the increasing cumulative irradiation fluence; meanwhile, the dislocation pattern also changes into a pattern where the dislocation line occupies the main direction, accompanied by an admixture of dislocation lines and groups. Figure 4 shows the microstructure of the 600 °C-tempered AISI 420 stainless steel after proton irradiation. From Figure 4, it can be seen that the variations in the dislocations in the microstructure of the sample without proton irradiation and with proton irradiation are similar to those in Figure 3. There are also a small number of dislocations in the non-irradiated specimen, as shown in Figure 4a. As the cumulative proton irradiation fluence increases, the dislocation density in the 600 °C-tempered specimen also increases and becomes significantly larger than that of the annealed specimen under the same conditions. It can be seen from Figure 4b–d that the increases are significantly smaller than those in the annealed specimen. When the tempering temperature is 700 °C (see Figure 5), the grains in the AISI 420 stainless steel structure become larger than those in the 600 °C-tempered specimen, and smaller than those in the annealed specimen. The variation trend for the effect of heat-treatment is mainly reflected by the variation in the grain size. It can be clearly seen from Figure 3, Figure 4 and Figure 5 that the grain size of the annealed steel is the largest, because the slow cooling process allows the grains to have enough time to grow. The grain size in the 600 °C-tempered steel is the smallest. This can be ascribed to recrystallization during the tempering process, which refines the grains after the high-temperature quenching process. As the tempering temperature rises to 700 °C, the grain size also increases, but it remains smaller than that of the annealed steel. The effect of proton irradiation leads to new vacancies, interstitial atoms, and other defects occurring in an extremely thin region near the material’s surface layer. The increase in defects is mainly reflected by an increase in the dislocation density and a change in the dislocation pattern. As the cumulative proton irradiation fluence increases, the number of defects in the structure also increases near the surface layer of the material. Correspondingly, there are more obvious lattice distortions or a higher dislocation density. As a result, a radiation hardening effect acts upon the material’s surface, which increases the surface hardness of the material [18]. In addition, the presence of dislocations in samples that were not subject to proton irradiation was also observed in Figure 3a, Figure 4a and Figure 5a, thereby indicating that combined effect of the heat treatment process before proton irradiation caused a change in the microstructure of the irradiated surface layer [29].

### 3.2. Nano-Scratch Hardness for AISI 420 Stainless Steels with Different Heat-Treatments after Proton Irradiation

In order to verify the proton irradiation hardening effect of the AISI 420 stainless steel surface, the nano-scratch hardness was calculated according to Equation (1):
(1)$$H_{S}=2.31\frac{F_{N}}{d^{2}},$$
where *F*_N_ denotes the maximum normal force (μN) applied to the indenter, and d denotes the scratch width (μm). The nano-scratch hardness collected by the nano-scratch data acquisition system is shown in Figure 6. For different heat treatments and the action of three loads, Figure 6 shows a bar graph describing the relationship between the nano-scratch hardness of the AISI 420 stainless steel and the cumulative proton irradiation fluences. It can be seen from Figure 6a–c that when the heat-treated state and the cumulative fluence of proton irradiation are constant, the nano-scratch hardness of the AISI 420 stainless steel decreases first, and then it increases with a higher load.

When the heat treatment state and the load are constant, the nano-scratch hardness of the AISI 420 stainless steel increases with a higher cumulative fluence of proton irradiation. This trend has a very good correspondence with the microstructural evolution shown in Figure 3, Figure 4 and Figure 5.

The evolution of the microstructure in the AISI 420 stainless steel was analyzed by TEM, and the hardness was analyzed via the nano-scratch tests. It was found that an increase in the cumulative proton irradiation fluence could lead to the formation of vacancies or interstitial atoms on the surface layer of the AISI 420 stainless steel. This could induce a higher dislocation density in the microstructure, and result in a surface hardening effect on the material. By analyzing the nano-scratch hardness of AISI 420 stainless steel in the same heat-treated state, it was found that the hardness increases with a higher cumulative proton irradiation fluence, indicating the occurrence of a proton radiation-induced hardening effect.

### 3.3. Nanoscale Friction Coefficients of Differently Heat-Treated AISI 420 Stainless Steels after Irradiation

Figure 7, Figure 8 and Figure 9 show the variation of the nanoscale friction coefficient of annealed, 600 °C-tempered, and 700 °C-tempered AISI 420 stainless steel specimens. The samples were subject to proton irradiation with different cumulative fluences and under a pressure load of 2000 μN, 3000 μN, or 4000 μN. The variation of the friction coefficients can be seen in Figure 7, Figure 8 and Figure 9. The first 17 seconds is the pre-scratch stage, where the nano-scratch indenter barely touches the material’s surface. The friction coefficient sharply rises at around the 17th second, indicating that the friction coefficient of the indenter begins to increase under the material’s resistance to deformation. The period between the 17th and 45th second is the stable frictional wear stage, in which the friction coefficient is affected by the coupled effects of the pressure load and the cumulative proton irradiation fluence. When the pressure load is constant, the variation of the friction coefficient decreases with increasing total proton irradiation fluence. When the cumulative fluence is constant, the friction coefficient increases with a higher pressure load, but the magnitude of increase is not large.

For ease of analysis, we present the mean friction coefficient in Table 4, which was calculated based on the data in Figure 7, Figure 8 and Figure 9. The mean value is obtained by dividing the sum of the maximum and minimum values of the friction coefficient curve by two, during the stable friction stage. For the 600 °C-tempered steel in Table 4 for instance, when the scratch load is 4000 μN, the average nanoscale friction coefficient corresponding to the three cumulative fluences 2 × 10^14^ p/cm^2^, 2 × 10^15^ p/cm^2^, and 2 × 10^16^ p/cm^2^ were 2.8%, 9.4% and 10.4%, respectively, which were lower than those of the non-irradiated steel. The nanoscale friction coefficients under other conditions also showed the same trend, indicating that the surface hardening effects induced by proton irradiation can reduce the nanoscale friction coefficient. It can be seen from Table 4 that the variation in the mean friction coefficient is related to the heat treatment temperature, cumulative proton irradiation fluence, and the load on the nano-scratch test.

When the load of the nano-scratch is constant, the friction coefficients of the AISI 420 stainless steel without proton irradiation or with proton irradiation can both be characterized accordingly: the friction coefficient of the annealed specimen is larger, while the friction coefficient of the 600 °C-tempered specimen is smaller; the friction coefficient of the 700 °C-tempered specimen is slightly larger than that of 600 °C-tempered specimen, but it is smaller than that of the annealed specimen. The variation of the friction coefficients may be due to the variations in the grain size of the AISI 420 stainless steel structures, due to the different heat treatment temperatures, and the surface hardening induced by proton radiation.

Generally speaking, the slow cooling process provides the grains of the annealed AISI 420 stainless steel specimen sufficient growth time, so that its grain size is larger than that of the 600 °C-tempered sample.

### 3.4. Deformation of a Nano-Scratch with Nano-Scratching of Different Heat-Treated AISI 420 Stainless Steels after Proton Irradiation 

The AISI 420 stainless steel was annealed or tempered at 600 and 700 °C, and subjected to proton irradiation with different cumulative fluences. Figure 10 illustrates the cross-sectional profile curve of the nano-scratch, using AFM attached to the nanomechanical tester. The red dashed lines in Figure 10a–c indicate the positions of samples’ surfaces; the curve below the red line indicates the nano-scratch depth, while the curve above the red line indicates the ridge height of the nano-scratch. It can be seen from Figure 10 that under the intrusion and pushing of the indenter, the specimen’s surface was plastically deformed; ridges appeared on both sides of the scratch, and grooves appeared in the middle. The groove depth and the ridge height on both sides can characterize the degree of plastic deformation and wear.

It can be seen from Figure 10 that the deformation and depth of the nano-scratches on the AISI 420 steel are closely related to the heat-treated states and the cumulative proton irradiation fluence. When the heat-treated states are identical, the plastic deformation of the scratch becomes larger as the load increases. When the load is constant, the scratch’s plastic deformation decreases as the cumulative fluence of proton irradiation increases. It can also be seen from Figure 10 that the nano-scratch depth, width, and height of the annealed specimen (Figure 10a) and 700 °C-tempered specimen (Figure 10c) are both larger than those of the 600 °C-tempered specimen (Figure 10b), which can be related to the microstructure in the different heat-treated states. The variation in the microstructural morphology also exhibits a good correlation with the microstructure evolution shown in Figure 3, Figure 4 and Figure 5.

### 3.5. Wear Volume with Nano-Scratching of Different Heat-Treated AISI 420 Stainless Steels after Proton Irradiation

According to the profile curve of the nano-scratch’s sectional deformation in Figure 10, the wear volume of the nano-scratch is closely related to the scratch’s length and depth. In order to evaluate the wear volume of the nano-scratch, the relevant parameters of the nano-scratch cross-section are given by Figure 11, and the wear volume was calculated. Figure 11a illustrates the scanning probe microscopy (SPM) images of the nano-scratch on the 600 °C-tempered AISI 420 stainless steel after proton irradiation with a fluence of 2 × 10^15^ p/cm^2^, and under loads of 4000 μN, 3000 μN, and 2000 μN. Figure 10b is a plot showing the depth variation of the nano-scratch cross section under 4000 μN. The wear volume of the nano-scratch caused by the Berkovich indenter can be calculated by Equation (2) [30]:
(2)$$V=\int_{-l/2}^{l/2}\frac{1}{2}\left[\tan\phi +\tan\left(\theta -\phi \right)\right]h^{2}dx=\int_{-l/2}^{l/2}Ch^{2}dx$$
where *h* denotes the depth of the scratch (nm); *x* denotes the spacing between the scratches (μm); *l* denotes the scratch length (μm); *C* denotes the cross-sectional area factor; *θ* denotes the intersection angle between the two sides of the scratch groove; *ϕ* denotes the angle between one side of the scratch’s groove and the normal direction. The parameters *h*, *l*, *θ*, and *ϕ* in Equation (2) can be obtained from the nano-scratch cross-sectional profile curve of Figure 11b.

Sribalaji et al. [31] summarized the simplified Equation (3) to calculate the wear volume of the nano-scratch, using a Berkovich indenter:(3)$$V=\frac{1}{2}\cos(70.3)h^{2}l$$
where *V* denotes the wear volume (μm^3^) of the nano-scratch, *h* denotes the depth (nm) of the nano-scratch groove, and *l* denotes the length (μm) of the nano-scratch. Based on the maximum depth and the scratch length (15 μm) measured from cross-sectional profile curve in Figure 10, the wear volume calculated according to Equation (3) and the measured maximum depth of the nano-scratch are listed in Table 5.

## 4. Discussion

Since the penetration depth of 160 KeV protons is of the order of only 0.9 [32], the friction coefficient and the surface tribological and wear properties were investigated by using a scratch test with a nano-indenter, which affects material depths of approximately 150 nm, as shown in Table 5. The modifications of the microstructure shown in Figure 3, Figure 4 and Figure 5 are also representative of irradiation, since the TEM foils are typically taken at a depth of less than 200 nm underneath the surface, which corresponds to the maximum material transparency of 200 KeV electrons.

The coefficient of friction has an important effect and influence on the wear behavior of two mutually touching moving surfaces. The coefficient of friction is affected by many factors, among which the degree of contact surface roughness and the nature of the material itself have a great influence on the friction coefficient.

When the tempering temperature is raised to 700 °C, the grain size also increases, due to the higher temperature. From the above analysis, the variation in the nano-scratch friction coefficient of the AISI 420 stainless steel is related to the grain size of the microstructure during different heat treatments. When the type of heat treatment is the same, the variation in the nanoscale friction coefficient is due to the coupled action of the nano-scratch load, and the cumulative fluence of the proton irradiation. When the cumulative fluence is constant, the friction coefficient increases with increasing load. When the load is constant, the friction coefficient decreases with increasing cumulative fluence. This is due to the surface hardening effect induced by proton irradiation. With the increase in the cumulative proton irradiation fluence, the surface hardness of the AISI 420 stainless steel also increases. During the test with the nano-scratch indenter, since the surface hardness was high, plastic deformation did not easily occur; hence, the frictional resistance decreased and manifested as a decreasing nanoscale friction coefficient. In addition, the different microstructures and precipitates formed after various heat treatments also have an important effect on the friction coefficient. The annealed specimen is composed of ferrite and large carbides. Because of the soft ferrite, the annealed specimen has a low resistibility to friction and wear, and a relatively large friction coefficient. The microstructural characteristics of specimens tempered at 600 °C and 700 °C are primarily martensite and a few carbides. Due to the high strength and hardness of martensite, the resistance to friction and wear of tempered specimens is higher than that of annealed specimens, and the corresponding friction coefficient is also reduced. Although carbides should also play an important role in friction and wear, their amount is too small to obviously influence the friction behavior in this work. From the above analysis, it can be seen that the change of nano-friction coefficient has a good correspondence with the evolution of the microstructure in Figure 3, Figure 4 and Figure 5, and the change of surface hardness in Figure 6.

Wear damage is the most severe failure mode with relative motion mechanisms. The main parameters that are evaluated are wear volume and wear rate. The wear volume represents the change in the volume of the worn sample, and the wear rate characterizes the rate at which the sample wears.

It can be seen from the data in Table 5 that when the loads were 4000 μN and 3000 μN, the annealed sample had the largest scratch depth for the different cumulative fluences of proton irradiation, and the scratch depth of the 600 °C-tempered steel was the smallest value. The scratch depth for the 700 °C-tempered steel was larger than that of the 600 °C-tempered steel, but smaller than that of the annealed steel. Under a load of 2000 μN, the values of the nano-scratch depth were slightly discrete. This may be due to the fact that the Berkovich indenter is more sensitive to defects or uneven areas inside the material during the scratching process under lower loads; however, the overall variation trend was basically the same as that under a higher load. That is, when the heat-treated state is constant, the nano-scratch depth decreases with a higher cumulative proton irradiation fluence. When the cumulative fluence is constant, the nano-scratch depth increases with the load on the indenter. As for the wear volume in Table 5, its variation is consistent with that of the nano-scratch depth, since the wear volume relies on the scratch depth. In addition to the friction coefficient and the wear volume, the wear rate is an important parameter in evaluating the tribological and wear properties. The wear rate can be used to characterize the rate of the nano-scratching process. When the pressure load, nano-scratch length, and wear volume are known, the wear rate *K* can be calculated by Equation (4):(4)$$K=\frac{V}{P\times L}$$
where *P* denotes the pressure load, *L* denotes the length of the nano-scratch, and *V* denotes the wear volume. Under a cumulative fluence of 2 × 10^16^ p/cm^2^ and with different loads, the wear rates of the 600-tempered AISI 420 stainless steel in Table 5 can be calculated according to Equation (4). After proton irradiation with a cumulative fluence of 2 × 10^16^ p/cm^2^, and under loads of 2000 μN, 3000 μN, and 4000 μN, the corresponding wear rate reduces by 45.58%, 4.92%, and 16.98% relative to that of non-irradiated stainless steel. The reduction in the wear rate of the nano-scratch indicates that the wear resistance during the scratching process was enhanced. This can be attributed to the proton irradiation hardening effect, and a lower friction coefficient.

## 5. Conclusions

In this work, the microstructure evolution and nanotribological properties of different heat-treated AISI 420 stainless steels after proton irradiation with three different cumulative fluences was studied. Heat treatment procedure at 680 °C annealing, tempering at 600 and 700 °C, and proton irradiation with different cumulative fluences can cause variations in the microstructure and grain size of AISI 420 stainless steel. The grain size of the annealed specimen was the largest, while the grain size of the tempered specimen, at 600 °C, was the smallest; the grain size of the 700 °C-tempered specimen was larger than that of the 600 °C-tempered specimen, but still smaller than the annealed sample. The radiation-hardening effect induced by proton irradiation increases the surface hardness. The nano-scratch hardness of the AISI 420 stainless steel after the heat treatments was also investigated. It was found that the hardness of the nano-scratch was higher as the cumulative proton irradiation fluence increased, indicating a proton radiation-induced hardening effect.

The heat treatment and proton irradiation have important effects on the tribological and wear properties of AISI 420 stainless steel. At a nano-scratch load of 4000 μN, for fluences of 2 × 10^14^ p/cm^2^, 2 × 10^15^ p/cm^2^, and 2 × 10^16^ p/cm^2^, the nanoscale friction coefficients were reduced by 2.8%, 9.4%, and 10.4%, respectively, for the 600 °C-tempered specimen, compared to the non-irradiated specimen. When the fluence was fixed at 2 × 10^16^ p/cm^2^, the wear rates of the specimen under 2000 μN, 3000 μN, and 4000 μN presented reductions of 45.58%, 4.92%, and 16.98%, compared with those of the non-irradiated specimen.

## Figures and Tables

**Figure 1 materials-12-01736-f001:**
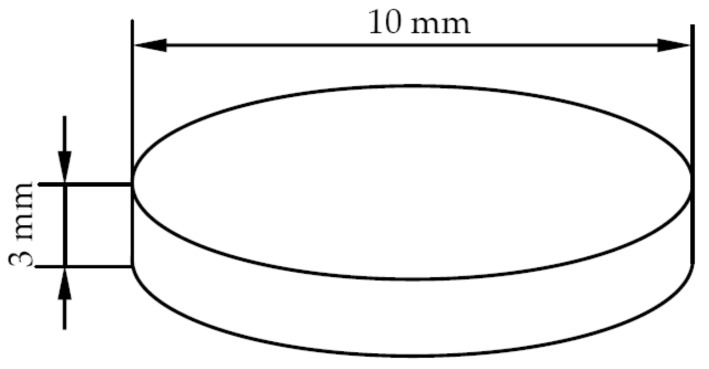
The proton irradiation sample size.

**Figure 2 materials-12-01736-f002:**
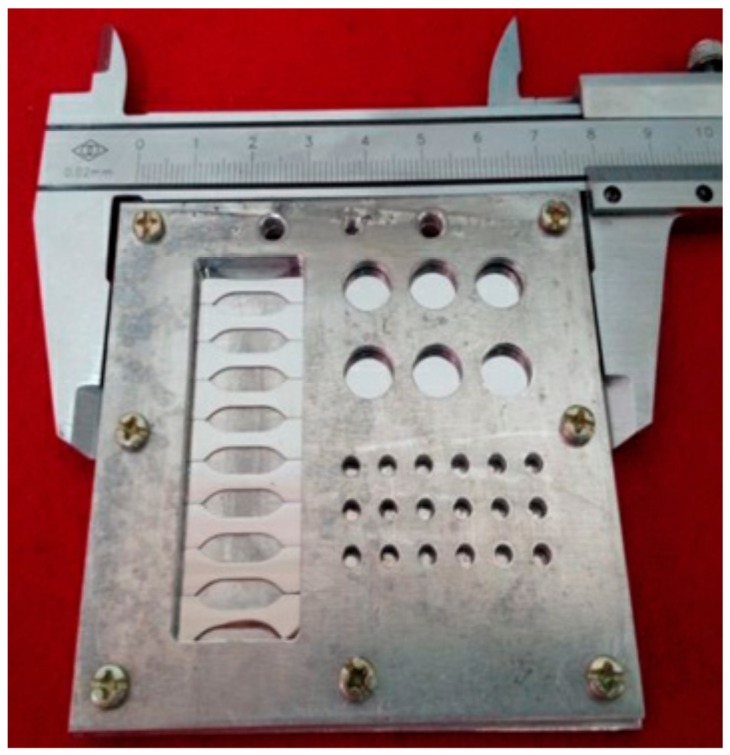
Clamping fixture holding the specimens during proton irradiation.

**Figure 3 materials-12-01736-f003:**
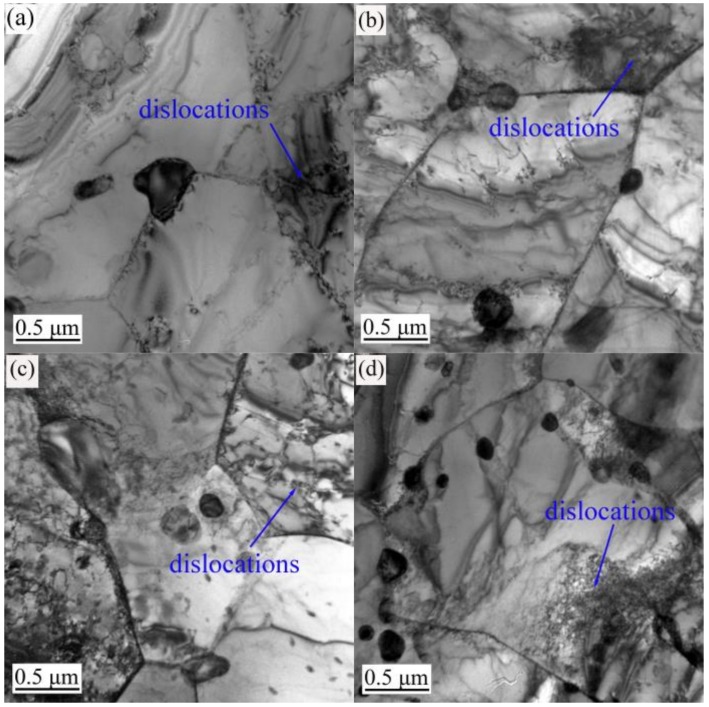
Transmission electron microscopy (TEM) observations of the annealed AISI 420 steel after proton radiation with different cumulative charges (**a**) 0 p/cm^2^; (**b**) 2 × 10^14^ p/cm^2^; (**c**) 2 × 10^15^ p/cm^2^; (**d**) 2 × 10^16^ p/cm^2^.

**Figure 4 materials-12-01736-f004:**
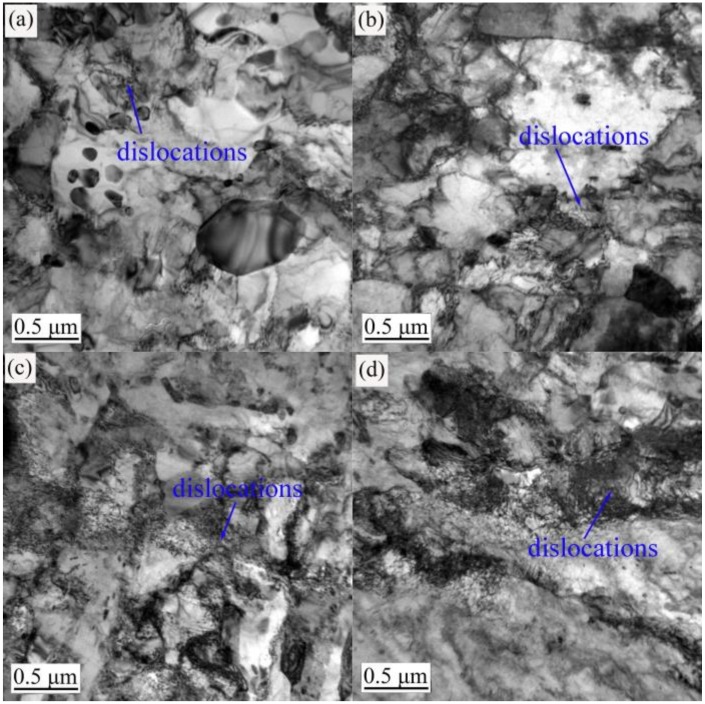
TEM observations of the 600 °C-tempered AISI 420 steel after proton radiation with different cumulative charges (**a**) 0 p/cm^2^; (**b**) 2 × 10^14^ p/cm^2^; (**c**) 2 × 10^15^ p/cm^2^; (**d**) 2 × 10^16^ p/cm^2^.

**Figure 5 materials-12-01736-f005:**
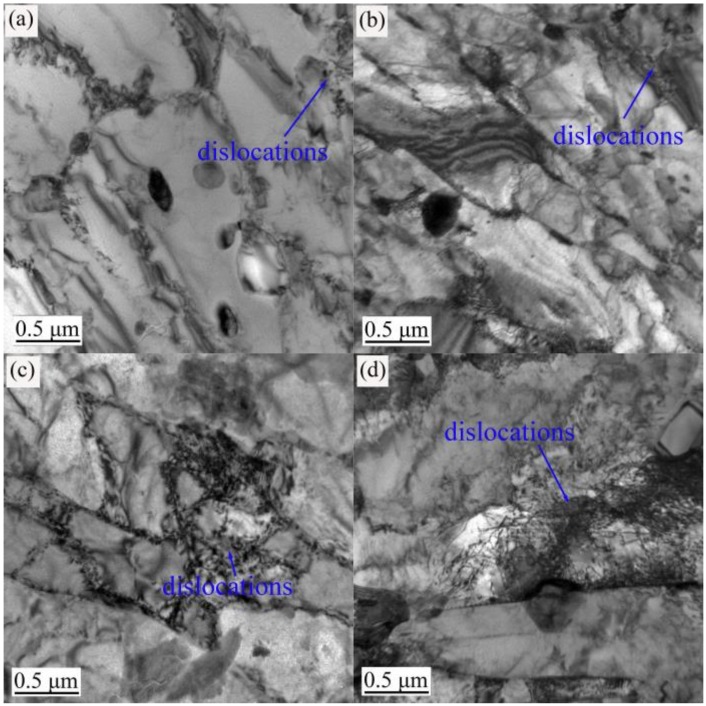
TEM observations of the 700 °C-tempered AISI 420 steel after proton radiation with different cumulative charges (**a**) 0 p/cm^2^; (**b**) 2 × 10^14^ p/cm^2^; (**c**) 2 × 10^15^ p/cm^2^; (**d**) 2 × 10^16^ p/cm^2^.

**Figure 6 materials-12-01736-f006:**
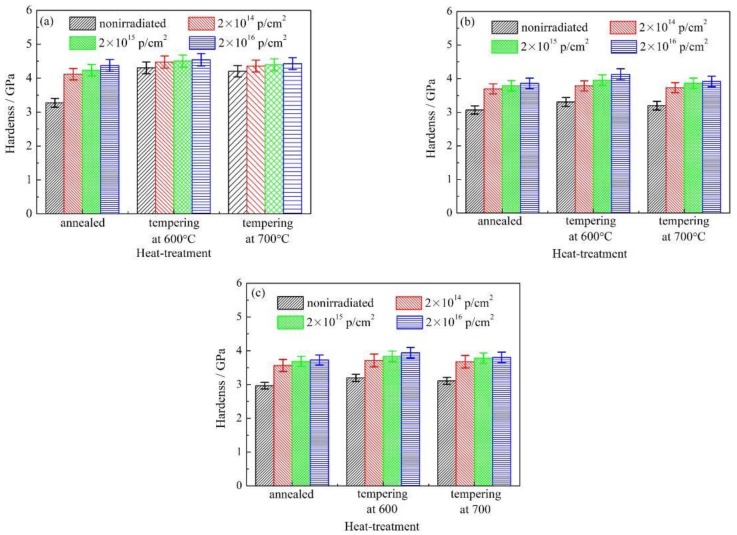
Nano-hardness of AISI 420 steel after proton radiation with different cumulative charges. (**a**) 2000 μN; (**b**) 3000 μN; (**c**) 4000 μN.

**Figure 7 materials-12-01736-f007:**
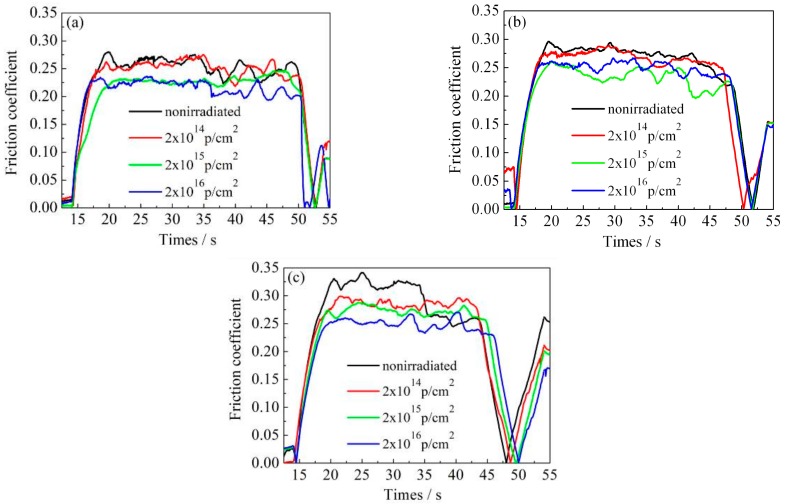
Friction coefficient variations of annealed AISI 420 steels after proton radiation with different cumulative charges. (**a**) 2000 μN; (**b**) 3000 μN; (**c**) 4000 μN.

**Figure 8 materials-12-01736-f008:**
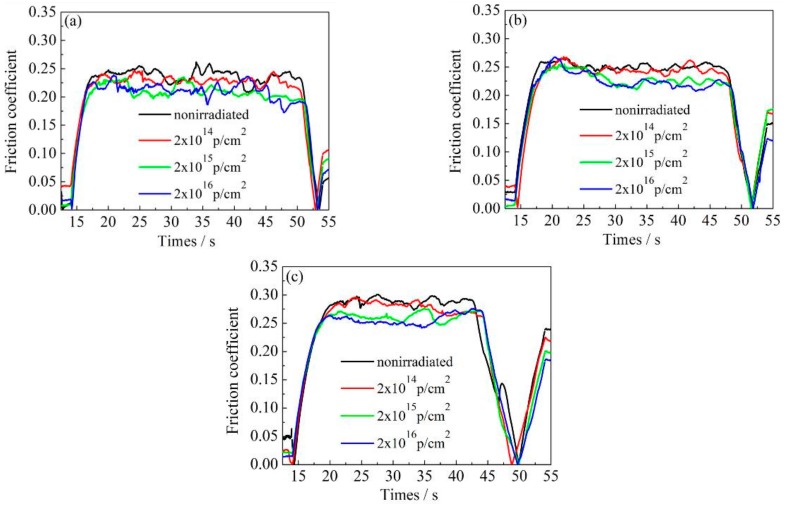
Friction coefficient variations of 600 °C-tempered AISI 420 steels after proton radiation with different cumulative charges. (**a**) 2000 μN; (**b**) 3000 μN; (**c**) 4000 μN.

**Figure 9 materials-12-01736-f009:**
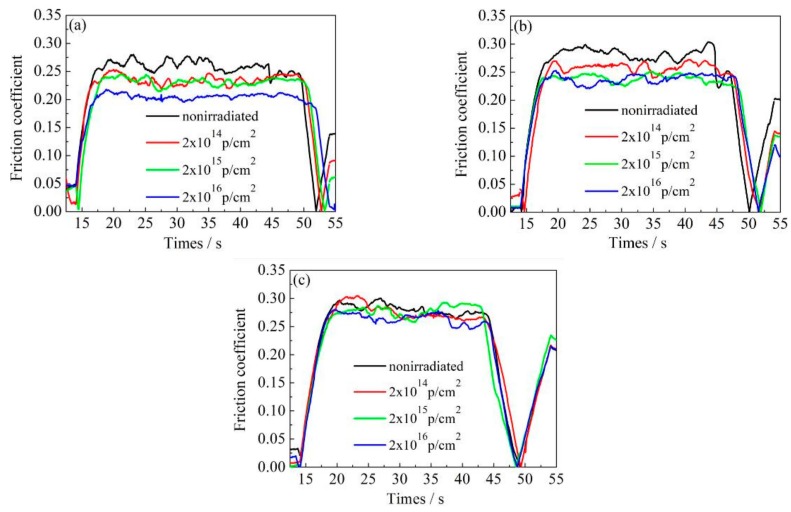
Friction coefficient variations of 700 °C-tempered AISI 420 steels after proton radiation with different cumulative charges. (**a**) 2000 μN; (**b**) 3000 μN; (**c**) 4000 μN.

**Figure 10 materials-12-01736-f010:**
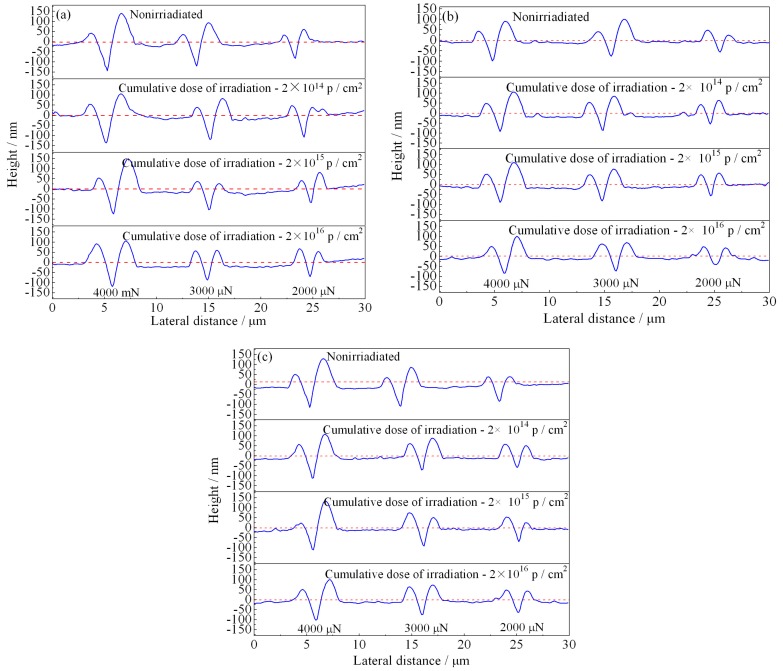
Wear profile of heat-treated AISI 420 steel after proton radiation with different cumulative charges. (**a**) Annealing state; (**b**) 600 °C; (**c**) 700 °C.

**Figure 11 materials-12-01736-f011:**
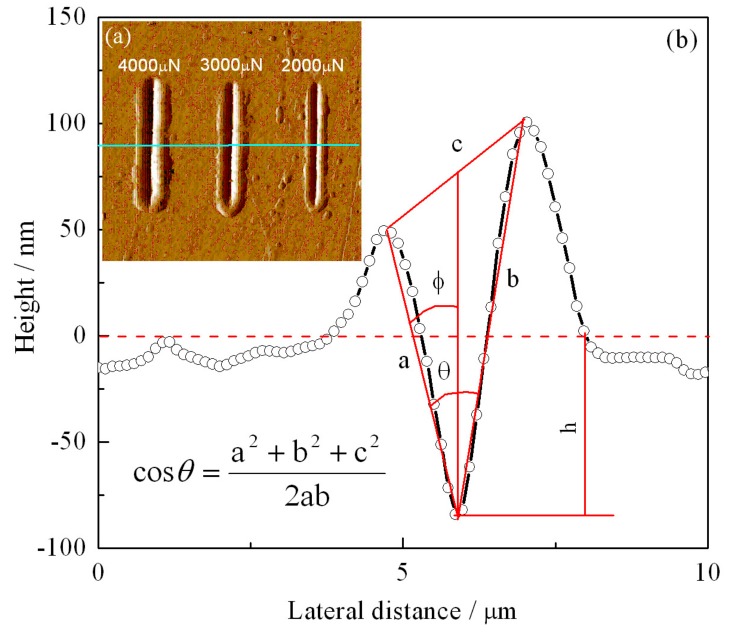
(**a**) Nano-scratch of scanning probe topography and (**b**) cross-section of the scratch depth of 600 °C-tempered, cumulative proton irradiation flux 2 × 10^15^ p/cm^2^ AISI 420 steel under a 4000 μN normal load condition.

**Table 1 materials-12-01736-t001:** The chemical composition of AISI 420 stainless steel (wt %).

Material	C	Mn	Si	S	P	Ni	Cr	Cu	Fe
AISI 420	0.21	0.27	0.53	0.009	0.025	0.18	12.23	-	86.546

**Table 2 materials-12-01736-t002:** The heat-treatment of AISI 420 stainless steel.

Material	Size (mm)	Quenching Temperature (°C)	Holding Time (h)	Cooling Method	Anneal Temperature (°C)	Tempering Temperature (°C)	Holding Time (h)	Cooling Method
AISI 420	D = 20	-	-	-	680	-	-	Cooling with furnace to 350 °C Air cooling to room temperature
		980	1	oil cooling	-	600	2.5	Oil cooling
					-	700	2.5	Oil cooling

**Table 3 materials-12-01736-t003:** The proton irradiation test parameters.

Radiation Environment (Pa)	Radiation Energy (KeV)	Radiant Flux (protons·cm^−2^·s^−1^)	Radiation Temperature(°C)	Radiation Accumulative Proton (protons/cm^2^)
10^−3^–10^−4^	160	1.25 × 10^11^	Room temperature	2 × 10^14^
				2 × 10^15^
				2 × 10^16^

**Table 4 materials-12-01736-t004:** Average nano-friction coefficients of AISI 420 stainless steels after proton radiation with different cumulative charges.

Loads (μN)	Cumulative Protons (p/cm^2^)	Coefficient of Friction					
		Annealing Status	Reduced %	Tempering at 600 °C	Reduced %	Tempering at 700 °C	Reduced %
2000	0	0.253 ± 0.028	0	0.235 ± 0.026	0	0.260 ± 0.021	0
	2 × 10^14^	0.247 ± 0.028	2.3	0.230 ± 0.017	2.1	0.236 ± 0.017	9.2
	2 × 10^15^	0.232 ± 0.014	8.3	0.214 ± 0.020	8.9	0.229 ± 0.016	11.9
	2 × 10^16^	0.216 ± 0.021	14.6	0.211 ± 0.027	10.2	0.207 ± 0.012	20.4
3000	0	0.266 ± 0.030	0	0.254 ± 0.012	0	0.262 ± 0.041	0
	2 × 10^14^	0.265 ± 0.028	1.02	0.250 ± 0.016	1.6	0.256 ± 0.017	2.3
	2 × 10^15^	0.250 ± 0.018	6	0.233 ± 0.023	8.3	0.238 ± 0.013	9.2
	2 × 10^16^	0.229 ± 0.032	13.9	0.238 ± 0.031	6.3	0.236 ± 0.015	9.9
4000	0	0.293 ± 0.049	0	0.288 ± 0.014	0	0.283 ± 0.016	0
	2 × 10^14^	0.286 ± 0.013	2.3	0.280 ± 0.016	2.8	0.281 ± 0.024	0.7
	2 × 10^15^	0.274 ± 0.015	6.5	0.261 ± 0.014	9.4	0.276 ± 0.017	2.5
	2 × 10^16^	0.251 ± 0.020	14.3	0.258 ± 0.017	10.4	0.262 ± 0.018	7.4

**Table 5 materials-12-01736-t005:** The wear depth and wear volume of AISI 420 stainless steel after proton radiation with different cumulative charges.

Loads (μN)	Cumulative Protons (p/cm^−2^)	Annealing Status		Tempering at 600 °C		Tempering at 700 °C	
		Maximum Depth (nm)	Wear Volume (μm^3^)	Maximum Depth (nm)	Wear Volume (μm^3^)	Maximum Depth (nm)	Wear Volume (μm^3^)
**2000**	0	82.15	2.08 × 10^−3^	56.38	1.43 × 10^−3^	84.48	2.14 × 10^−3^
	2 × 10^14^	103.75	2.62 × 10^−3^	53.83	1.36 × 10^−3^	57.56	1.46 × 10^−3^
	2 × 10^15^	64.95	1.64 × 10^−3^	56.16	1.42 × 10^−3^	69.05	1.75 × 10^−3^
	2 × 10^16^	69.82	1.77 × 10^−3^	40.00	1.01 × 10^−3^	62.91	1.59 × 10^−3^
3000	0	123.43	3.12 × 10^−3^	75.89	1.92 × 10^−3^	107.79	2.73 × 10^−3^
	2 × 10^14^	120.03	3.03 × 10^−3^	85.91	2.17 × 10^−3^	71.37	1.80 × 10^−3^
	2 × 10^15^	103.72	2.62 × 10^−3^	82.85	2.09 × 10^−3^	91.82	2.32 × 10^−3^
	2 × 10^16^	87.48	2.21 × 10^−3^	72.49	1.83 × 10^−3^	74.99	1.90 × 10^−3^
4000	0	143.61	3.63 × 10^−3^	98.20	2.48 × 10^−3^	113.92	2.88 × 10^−3^
	2 × 10^14^	136.35	3.45 × 10^−3^	91.07	2.30 × 10^−3^	111.79	2.82 × 10^−3^
	2 × 10^15^	123.71	3.13 × 10^−3^	88.99	2.25 × 10^−3^	109.62	2.77 × 10^−3^
	2 × 10^16^	119.24	3.01 × 10^−3^	84.03	2.12 × 10^−3^	102.36	2.59 × 10^−3^

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
