# Peer review of "Microstructure Evolution and Nanotribological Properties of Different Heat-Treated AISI 420 Stainless Steels after Proton Irradiation"

_materials, 2019, doi:10.3390/ma12111736_

Reviewer 1 Report

Dear authors,

Abstract: mention that your protons are low energy. Line 12-13: Mention also that the microstructure was studied close to the surface.

Introduction: You should add a remark for the non-specialist that protons are charge particles and have a penetration into materials dependent on their energy. High energy protons modify the bulk properties, low energy protons only the surface.

Materials and methods: The description of the heat treatments in your Table 2 is not clear. Please describe the heat treatments in text. D=20?

Your material was homogenized for 1hr at 980°C then quenched in oil? then was annealed at 680°C for how long?

The tempering heat treatments were done subsequently on the annealed material, for 2.5 hrs? then oil cooling?

It would be also helpful to give the basic mechanical properties ,yield stress and ductility after the heat treatments

Line 95: Your irradiation technique is not described, you only give some parameters.

You need to precisely describe the irradiation technique and add a reference.

Irradiation temperature is not mentioned! Irradiation temperature is a critical parameter for the formation of the irradiation defects.

Results :Line 120-150: You mention a proton energy of 160 KeV, at that energy the proton range should not exceed 1 micrometer. It is difficult to admit that the irradiation damage can affect the volume typically observed in a TEM ( a few micrometers).

So the modifications you are describing in Figure 2-4 are probably not due to the protons only!

Line 144-145: To understand your statement on the grain size one should understand clearly the heat treatments received by the material, 680 is very close to 700°C, do you have enough observations to derive a precise value?

For figures , you write Figure.1 etc.. remove the dot (all paper) !

3.2 Nano- hardness for AISI stainless steels with different heat treatments:

In the title, typing error hardness.

Your statement on line 182 is supported by the mechanical results mainly.

Line 210: 2.3 should read 2.8?

3.3 Nanoscale friction coefficient...: the explanations from line 240 to 253 should be in the discussion

Discussion: You are presenting many new measurements in 4.1 and 4.2 which are shown in Fig.         9 and Table 5. These results should be presented in the Results section.

Figure 9 presents measurements by AFM. Please describe the AFM in section 2.

 The profiles shown in Figure 9 and  qualitatively in Figure 10 seem to indicate that         the material was plastically deformed and pushed to the sides to form ridges.             Qualitatively the cross surface of these ridges appears to be larger than the             scratch formed. Is it correct to speak from weight loss and worn off wear volume.             Do you have an evidence that material is effectively removed?

 Conclusions: Line 337. Your statement on heat treatment and proton irradiation is     too general and not specific to your results.

Author Response

Thanks for your’ useful comments and suggestions for our manuscript “Microstructure evolution and nanotribological properties of differently heat-treated AISI 420 stainless steelafter proton irradiation” (No.: 499025). We studied, revised and replied the suggestions carefully one by one. The modifies are marked using red font for comparison. The English have also been checked and modified by some experts using English as the first language.

Reviewer 2 Report

  how many determinations were made for nano-hardness?

the results in Figure 5 (line 178) represent the average of nano-harness ?

line 195 Fiure 7;

line 322 in eq. 4, appear L' but in presentation appear L

Author Response

Point 1:how many determinations were made for nano-hardness?

Response 1: We performed 5 repeated measurements on nano-hardness.

Point 2:the results in Figure 5 (line 178) represent the average of nano-harness ?

Response 2: Yes.

Point 3:line 195 Fiure7;line322 in eq. 4, appear L' but in presentation appear L

Response 3: We have corrected it in Line 355.

Reviewer 3 Report

The paper is well organized and written. I just suggest adding some new references. 

Author Response

Point 1:The paper is well organized and written. I just suggest adding some new references.

Response 1: Four new references has added in to manuscript in Line 60[25], 69[26], 104[27] and 182[29].

Round  2

Reviewer 1 Report

Thanks for the revision. Please consider following improvements/remarks. Changes mandatory.

Still a few mistakes to correct!

1. Abstract: The authors should point out from the beginning that they are using low  energy protons: Please change first sequence line 11 to:

In this paper,  low energy proton irradiation experiments with different …etc...

2. Materials and methods: Please improve the text as follows:  The material received two kinds of heat treatment in this study: the first one is an annealing process in which the AISI 420 stainless steel round bar with a diameter of 20 mm was raised to 680°C at a heating rate of 10°C/min, then cooled to 350°C in the furnace and finally air  cooled to room temperature. The second one is a tempering process  in which the AISI 420 stainless steel round bar with a diameter of 20 mm is homogenized at 980°C for 1 h then quenched to room temperature in oil and subsequently reheated to 600°C or 700°C for 2.5h then oil cooled to RT.

(according to your Table 3, if I understand it correctly!)

Please note that the first heat treatment will anneal the dislocations and have little influence on the precipitation since the time at 680°C is very small. The second treatment will dissolve completely the precipitation at 980°C  and rebuild it during the holds of 2.5 h duration. In the three different cases, you should have precipitates with three different distributions and therefore they will contribute to the results of the mechanical wear tests.

You never discuss the importance of the precipitates in your results. They can be seen in all your Figures 3,4 and 5 and they will contribute to the strength.

Line 106 to 109, in red : please change :

cleaning ( and polishing ?) treatment , the pellets are inserted into an irradiation specimen holder, as shown in Figure 2. The irradiation specimen holder is placed on the table of the irradiation target so that the exposed surface of the sample is perpendicular to the positive ion proton beam.

You should really tell if your pellets are polished because  otherwise it would be difficult to have a sufficient small homogeneous volume to have consistent TEM results..

Line 114: please change

Figure 2: Clamping fixture holding the specimens during proton irradiation

Line 115:

Table 3: Remove  radiation temperature, replace by irradiation temperature

below it, in the table, insert : room temperature

Line 119: replace substance by substrate

Captions of all figures: Please replace Protons by protons, the same in Table 4

4.Discussion

Line 317 to 319: Wrong statement!

Replace by: Since the penetration depth of 160 KeV protons is of the order of 0.9 micrometer only (add reference , for instance Janni), the friction coefficient and surface tribological and wear properties were investigated using the scratch test with a nano-identer which affects material depths of approximately 150 nm, as shown in Table 5. The modifications of the microstructure shown in Figures 3 to 5 are also representative of the irradiation since the TEM foils are taken underneath the surface typically less than 200 nm in depth,  which corresponds to the maximum material transparency of 200 KeV electrons.

Line337: change wear evaluation to wear damage !

Line 339: change proton-radiated hardening to proton irradiation hardening

Author Response

Dear reviewer

Thank you for your and reviewer’ useful comments and suggestions of our manuscript (No.499025). we have modified the manuscript accordingly, and detailed corrections are listed below point by point.

Kind regards, 
M.Z.Ma

Response to Reviewer 1 Comments

Point 1: 1. Abstract: The authors should point out from the beginning that they are using low energy protons: Please change first sequence line 11 to: In this paper, low energy proton irradiation experiments with different …etc...

In this paper, low energy proton irradiation experiments with different …etc..

Response 1: This sentence that “In this paper, low energy proton irradiation experiments with different …etc..” has added into the Abstract in Line 11.

Point 2: 2. Materials and methods: Please improve the text as follows: The material received two kinds of heat treatment in this study: the first one is an annealing process in which the AISI 420 stainless steel round bar with a diameter of 20 mm was raised to 680°C at a heating rate of 10°C/min, then cooled to 350°C in the furnace and finally air cooled to room temperature. The second one is a tempering process in which the AISI 420 stainless steel round bar with a diameter of 20 mm is homogenized at 980°C for 1 h then quenched to room temperature in oil and subsequently reheated to 600°C or 700°C for 2.5h then oil cooled to RT.

(according to your Table 3, if I understand it correctly!)

Please note that the first heat treatment will anneal the dislocations and have little influence on the precipitation since the time at 680°C is very small. The second treatment will dissolve completely the precipitation at 980°C and rebuild it during the holds of 2.5 h duration. In the three different cases, you should have precipitates with three different distributions and therefore they will contribute to the results of the mechanical wear tests.

You never discuss the importance of the precipitates in your results. They can be seen in all your Figures 3,4 and 5 and they will contribute to the strength.

Response 2: Additional discussion: In addition, the different microstructures and precipitates after various heat treatments also have an important effect on the friction coefficient. The annealed specimen is composed of ferrite and large carbides. Because of the soft ferrite, the annealed specimen has the low resist ability to friction and wear, and a relatively large friction coefficient. The microstructural characteristic of specimens tempered at 600 ℃ and 700 ℃ is major martensite and few carbides. Due to the high strength and hardness of martensite, the resistance to friction and wear of tempered specimens is higher than that of annealed, and the corresponding friction coefficient is also reduced. Although carbides should also play an important role in friction and wear, their amount is too small to influence the friction behavior obviously in this work. From the above analysis, it can be seen that the change of nano-friction coefficient has a good correspondence with the evolution of microstructure in Figuers 3, 4 and 5 and the change of surface hardness in Figuer 6. It has added into the Discussion in Line 337-348.

Point 3: Line 106 to 109, in red : please change :

cleaning ( and polishing ?) treatment , the pellets are inserted into an irradiation specimen holder, as shown in Figure 2. The irradiation specimen holder is placed on the table of the irradiation target so that the exposed surface of the sample is perpendicular to the positive ion proton beam.

Response 3: This sentence thatpolishing treatment, the pellets are inserted into an irradiation specimen holder, as shown in Figure 2. The irradiation specimen holder is placed on the table of the irradiation target so that the exposed surface of the sample is perpendicular to the positive ion proton beam. has added into the 2.2 Proton irradiation and nano-scratch test” in Line 106-108.

Point 4: Line 114: please change

Figure 2: Clamping fixture holding the specimens during proton irradiation

.

Response 4: It's changed. in Line 114.

Point 5: Line 115:

Table 3: Remove radiation temperature, replace by irradiation temperature below it, in the table, insert : room temperature

Response 5: Sorry for our unclear response for this issue before. Actually, the specimens were measured in vacuum and room temperature during the irradiation process. Therefore, there is no change for the irradiation temperature and table in text. Please understand that.

Point 6: Line 119: replace substance by substrate

Captions of all figures : Please replace Protons by protons, the same in Table 4

Response 6: Corrected it.

Point 7: 4.Discussion

Line 317 to 319: Wrong statement!

Replace by: Since the penetration depth of 160 KeV protons is of the order of 0.9 micrometer only (add reference, for instance Janni), the friction coefficient and surface tribological and wear properties were investigated using the scratch test with a nano-indenter which affects material depths of approximately 150 nm, as shown in Table 5. The modifications of the microstructure shown in Figures 3 to 5 are also representative of the irradiation since the TEM foils are taken underneath the surface typically less than 200 nm in depth, which corresponds to the maximum material transparency of 200 KeV electrons.

Response 7: This sentence that “Since the penetration depth of 160 KeV protons is of the order of 0.9 micrometer only [32], the friction coefficient and surface tribological and wear properties were investigated using the scratch test with a nano-indenter which affects material depths of approximately 150 nm, as shown in Table 5. The modifications of the microstructure shown in Figures 3 to 5 are also representative of the irradiation since the TEM foils are taken underneath the surface typically less than 200 nm in depth, which corresponds to the maximum material transparency of 200 KeV electrons.” has added into the Discussion in Line 317-322.

Added reference [32] in Line 476-478

Point 8: Line337: change wear evaluation to wear damage !

Response 8: In accordance with the requirements of the reviewers, Corrected it in Line 349.

Point 9: Line 339 (363): change proton-radiated hardening to proton irradiation hardening.

Response 9: In accordance with the requirements of the reviewers, Corrected it in Line 375